# Crosstalk between Oxidative Stress and Tauopathy

**DOI:** 10.3390/ijms20081959

**Published:** 2019-04-22

**Authors:** Md. Mamunul Haque, Dhiraj P. Murale, Yun Kyung Kim, Jun-Seok Lee

**Affiliations:** 1Molecular Recognition Research Center, Korea Institute of Science and Technology (KIST), Seoul 02792, Korea; mamun@chembiol.re.kr (M.M.H.); dhiraj.murale@chembiol.re.kr (D.P.M.); 2Bio-Med Division, KIST-School UST, Seoul 02792, Korea; 3Convergence Research Center for Diagnosis, Treatment and Care System of Dementia, Brain Science Institute (BSI), Korea Institute of Science and Technology (KIST), Seoul 02792, Korea

**Keywords:** oxidative stress, reactive oxygen species, tauopathy

## Abstract

Tauopathy is a collective term for neurodegenerative diseases associated with pathological modifications of tau protein. Tau modifications are mediated by many factors. Recently, reactive oxygen species (ROS) have attracted attention due to their upstream and downstream effects on tauopathy. In physiological conditions, healthy cells generate a moderate level of ROS for self-defense against foreign invaders. Imbalances between ROS and the anti-oxidation pathway cause an accumulation of excessive ROS. There is clear evidence that ROS directly promotes tau modifications in tauopathy. ROS is also highly upregulated in the patients’ brain of tauopathies, and anti-oxidants are currently prescribed as potential therapeutic agents for tauopathy. Thus, there is a clear connection between oxidative stress (OS) and tauopathies that needs to be studied in more detail. In this review, we will describe the chemical nature of ROS and their roles in tauopathy.

## 1. Tau Protein and its Pathogenicity

Tau protein is expressed abundantly in neurons as well as sparsely in non-neuronal cells like astrocytes and oligodendrocytes [1]. It is a microtubule-binding protein that gives microtubules’ integrity, which is critical for neuronal outgrowth [2,3,4]. It helps microtubules to anchor with other cytoskeletal filaments and organelles for structural support [5,6]. Microtubules are continuously assembled and disassembled in cells in a dynamic fashion, and this is maintained by the interaction between tau and the microtubule, which is tightly controlled by several factors. Modification of tau affects microtubule stabilization and other processes related to this protein [7]. Tau modification is promoted by post-translational modifications, conformational changes and the misfolding structure of tau. These modifications lead to the abnormal aggregation of tau into neurofibrillary tangle (NFT) structures. These NFTs accumulate in neurons, causing neuronal degeneration. Therefore, the formation of NFTs represents the significant pathological signatures in many neurodegenerative diseases classified as tauopathies [8]. The level of NFTs and tau modifications are correlated to the severity of the tauopathies, including Alzheimer’s disease (AD), Parkinson’s disease (PD), frontotemporal dementia (FTD), FTD with parkinsonism linked to chromosome-17 (FTDP-17), frontotemporal lobar degeneration (FTLD), Pick’s disease (PiD), progressive supranuclear palsy (PSP), corticobasal neurodegeneration (CBD), dementia pugilistica, etc. [9,10,11,12].

## 2. Causes of Tauopathies

Consequently, researchers have been studying the mechanism of tau pathogenesis. Tau is naturally a highly soluble protein, and it undergoes several modifications to become an aggregate [13,14]. The mechanisms for NFT formation from tau are still in debate today. Among them, aberrant posttranslational modifications (PTM) are the leading cause of this failure. In this regard, hyperphosphorylation, oxidation, proteolytic cleavage (truncation), acetylation, glycation, nitration, and conformational changes have been suggested to cause the neuro-pathogenicity of tau [13,15,16,17,18]. Apart from these hypotheses, imbalances in oxido-redox homeostasis, which produce reactive oxygen species (ROS), play significant roles in tauopathies.

## 3. Oxidative Stress and Its Relation to Tauopathies

ROS are oxygen-containing reactive molecules that are generated by oxidative stress (OS). A moderate level of ROS is critical in cellular defense mechanisms to fight against foreign subjects, and it triggers mitogen-activated protein kinase (MAPK) pathways to modulate cellular signaling (cell cycle, gene expression, cell survival and apoptosis) [19,20]. In normal physiological conditions, cells produce small amounts of ROS, and the levels of ROS are balanced by several antioxidant systems [21]. The imbalance between ROS generation and antioxidant defense causes the excessive accumulation of ROS, giving OS to the cells [21,22]. Thus, OS poses a significant threat to the brain, one of the most metabolically active organs, which is vulnerable to OS due to its high oxygen demand [23], abundance of the redox-active metals (iron or copper) [23], polyunsaturated fatty acids (substrates for lipid peroxidation) [24], and deficiency of the glutathione (GSH, an antioxidant to eliminate ROS) levels [25].

In age-related neurodegenerative diseases, balances between OS and antioxidant enzymes are distorted, resulting in various brain damages and neuronal death. There is increasing evidence that OS is one of the leading pathophysiological markers of tauopathies, and all of these findings suggests that there is a clear relationship between OS and the pathophysiology of tauopathies (Figure 1). Moreover, a series of studies have been focused on the elucidation of the mechanisms underlying ROS linked to tauopathies. However, it has not yet been fully understood whether OS is an early causal factor or a result of the cell injuries induced by tau modifications. Therefore, OS creates a scope for the development of therapeutic strategies for tauopathies. Here, we will discuss the cellular origin, reaction mechanism, and relation of ROS in tauopathies. 

There is clear evidence that OS contributes to neurological deterioration, as well as the oxidative destruction of nucleic acids, proteins, or lipids in the central nervous system (CNS) in tauopathies. OS mediated ROS production is involved in protein oxidation (glycoxidation) or lipid oxidation (lipoxidation) and forms stable advanced end products. These protein products are evident in NFTs in AD, whereas lipid products are present in neurofibrillary pathologies [26]. OS damages nucleic acid (DNA or RNA), and the guanine base is the most susceptible base for oxidative modifications (8-hydroxy-2’-deoxyguanosine, 8-OHdG or 8-hydroxy-2’-guanosine, 8-OHG). These damages by OS are another phenomenon that is associated with tauopathies [27].

AD is the best-known tauopathy, among others, with increasing prevalence. Accumulation of OS is directly linked to aging, and intimately related to AD [28]. In fact, OS is one of the first observable markers in AD progression, even before the appearance of amyloid beta (Aβ) accumulation [29]. Tau overexpressed cells show increased vulnerability to OS [30,31]. Furthermore, mice (P301S and P301L) with AD showed mitochondrial dysfunction, which is associated with increased ROS [32,33]. Several studies indicate that Aβ induces OS, where Aβ serves as a source of ROS to initiate lipid peroxidation [34]. Inversely, Aβ level is increased upon the stimulus of OS but diminishes with time [35]. These phenomena indicate that not only ROS modulates Aβ production, but also Aβ generates excessive ROS in cases of AD. Another report states that free radicals affected the nature and function of neural cells, both in AD and PD [28]. PD is another well-known tauopathy of which OS is the leading contributor. OS is believed to be the main etiology that leads to idiopathic and genetic causes of PD [28]. Alterations in the antioxidant molecules were evident in the early stages of PD [36].

## 4. Oxidative Stress and Reactive Oxygen Species

ROS are generated upon the reduction of O_2_, and they consist of free radicals and non-radical species. In particular, ROS that play critical roles in the biological system include superoxide anion radical (O_2_^−^ or O_2_•^−^), hydroperoxyl radical (HO_2_), hydrogen peroxide (H_2_O_2_), hydroxyl radical (•OH), ozone (O_3_), and singlet oxygen (^1^O_2_). Further, ROS also includes peroxyl (ROO•), alkoxyl (RO•), semiquinone (SQ•−), and carbonate (CO_3_•−) radicals, along with hypochlorous (HOCl), hypobromous (HOBr), and hypoiodous acids (HOI). Some of the reactive nitrogen species (RNS), which contain an oxygen atom, are also considered as one of these kinds, like the nitric oxide radical (NO or NO•), nitrogen dioxide radical (NO_2_•), nitrite (NO_2_−), and peroxynitrite (ONOO−) [37,38]. 

## 5. Major Sources of Oxidative Stress (OS)

The endogenous sources of OS are: (i) Hypoxia, NO and ONOO, which endorse the generation of ROS in mitochondrial electron transport chain (ETC) [37,39,40]; (ii) an increased level of misfolded proteins [41]; and (iii) intracellular enzymes that produce ROS as metabolic products of their enzymatic processes. These enzymes include NADPH oxidases, flavoenzyme ERO1, cytochrome p450, lipoxygenases, xanthine oxidase, nitric oxide synthases, etc. [37,38,40] Lastly, (iv) free metal cations (such as copper and iron ions) can convert superoxide or hydrogen peroxide into hydroxyl radicals through the Fenton reaction or the Haber–Weiss reaction [42]. A high level of intracellular Ca^2+^ is also one of the endogenous sources of OS [43,44]. The exogenous sources are ultraviolet light (^1^O_2_ generation), γ-irradiation (•OH generation) [45], chemical pollutants, including quinones, nitroaromatics, etc. These sources generate superoxides. It is also reported that smoke and air pollutants also contain ROS which potentially uptake into the body during the respiration process and induce OS [37,46].

## 6. Reactive Oxygen Species (ROS) Production in the Body

### 6.1. Mitochondria

Mitochondria are the major sub-cellular organelles for ROS production. ROS are produced as a byproduct of the electron transport chain reaction for adenosine triphosphate (ATP) generation. Loss of some mitochondrial proteins (cytochrome oxidase, COX; pyruvate dehydrogenase complex, PDHC; and α-ketoglutarate dehydrogenase complex, KGDHC) elevate the formation of ROS [47]. OS induces mitochondrial dysfunction by accumulating excessive ROS [30,48]. Increased oxidative damage may lead to mitochondrial DNA mutations, which has been reported to inhibit ATP production in AD patients [30]. In addition to mitochondria, endoplasmic reticulum, peroxisomes, and microglial cells are also potential sources of ROS.

### 6.2. Endoplasmic Reticulum

The endoplasmic reticulum (ER) is mainly responsible for protein folding and lipid biosynthesis. In the ER, ROS generation is induced by misfolded protein processing, depletion of the GSH, or through the breakage and formation of disulfide bonds [49].

### 6.3. Peroxisomes

Peroxisomes participate in many metabolic pathways that include fatty acid oxidation, amino acid catabolism, and phospholipid biosynthesis. They generate the majority of the H_2_O_2_ inside the body, which is balanced by catalases. This H_2_O_2_ level is increased when catalase is not working, or when peroxisomes are damaged. Peroxisomes have been shown to be depleted or damaged by tau aggregation in the rat primary hippocampal neurons, and N2a cells induce OS [31].

### 6.4. Microglia

Microglial cells express a high level of the glutathione peroxidase (GPx) antioxidant enzyme. They also produce ROS, either to eliminate pathogens or during neuroinflammatory processes. Recently, microglial cells were reported as ROS producers in tauopathies [11].

## 7. Chemical Properties of ROS

To gain a clearer perception of the ROS associated biological impact, one needs to understand the chemical properties of ROS.

### 7.1. Electronic Configuration of ROS

Molecular oxygen is paramagnetic species, having two electrons with a parallel spin in the π* orbital (Figure 2). Due to this kind of parallel spin, restriction is applied to an oxygen atom to take part in redox reactions. This kind of electronic configuration enables the oxygen atom to accept electrons during the redox reactions, making it unable to oxidize the biomolecules [38]. The molecular oxygen can be converted quite easily into ROS by either energy transfer or electron transfer. Among the ROS, energy transfer from oxygen results in the generation of singlet oxygen. As a result, singlet oxygen has paired electrons with opposite spin, which enhance the oxidizing ability when compared with molecular oxygen. By a one-electron reduction, O_2_ is converted into superoxide. Further, superoxide, with another one-electron reduction, converts into hydrogen peroxide which later converts into the hydroxyl radical by a one-electron reduction and finally the hydroxyl radical, upon reduction, converts into water. Hence, molecular oxygen can generate ROS by electron or energy transfer, making them more reactive than itself [50,51,52].

### 7.2. Reactivity Trends of ROS

To gain more in-depth knowledge into the reactivity of ROS, one should understand the thermodynamics of free radicals. To compare the trend of reactivity of the ROS, the role of reduction potential is vital. To plot the reactivity trend of ROS, one should consider the reduction potential of the one-electron reduction of molecular oxygen [38,50]. A few of the main features about the role of reduction potential and reactivity are listed as follows: (a) Reduction potential is the ability of an atom or molecule to acquire the electron; (b) oxidation is the loss of electron, whereas reduction is the gain of electron; and (c) from the above points, it is evident that atoms or molecules with high reduction potential have high reactivity as will be easy to reduce. Based on the above assumption, one can easily understand why molecular oxygen is a weak oxidant when compared with another ROS (Figure 3). Here, O_2_, with the reduction potential of −0.33 V, is a poor electron acceptor [53]. The superoxide radical also has limited reactivity when compared with its anionic charge, which makes it an electron-rich center [44]. Hydrogen peroxide is generated by the one-electron reduction of superoxide, which is also stable under physiological conditions, even though it has a +0.38 V reduction potential [52]. Among the ROS, the hydroxyl radical is the utmost powerful oxidizing agent, with a one-electron reduction potential of +2.33 V. According to the reduction potential, the ROS reactivity trend can be written as •OH > O_2_•^−^ > H_2_O_2_ > O_2_ [50,52].

## 8. ROS Generating Agents in Tauopathies

### 8.1. Tau Aggregation

Tau aggregate is one of the primary culprits in tauopathies. There is clear evidence among OS with tau hyperphosphorylation, polymerization, and toxicity in both animal models and patients [11]. OS produces oxidized fatty acids that stimulate in vitro tau polymerization [54]. Also, the overexpressed tau protein showed increased OS in N2a cultured cells [31]. Besides, rat cortical neurons from truncated tau expressing transgenic rats showed an increased level of ROS [55]. This evidence suggests that OS directly promotes tau aggregation, and inversely toxic tau species stimulate OS conditions in tauopathies.

### 8.2. Amyloid-Beta Aggregation

Amyloid-beta (Aβ) aggregation is another pathogenic event in tauopathies. Numerous studies have shown that OS promotes Aβ production by diminishing α-secretase activity and increasing β and γ-secretase activity [56]. The AD mouse model (Tg2576-APP-PS1) has enhanced OS due to increasing H_2_O_2_ levels and the peroxidation of proteins and lipids [57]. Oppositely, the accumulation of Aβ increases OS and leads to memory dysfunction in the AD mouse (Tg2576-APP) [58] and mitochondrial failure in the early stages of AD [30].

### 8.3. Metals

Copper (Cu), zinc (Zn) and iron (Fe) are the three most abundant metals in mammalian brains, which regulate many synaptic functions. Aberrant homeostasis of these metals, like their levels, mislocalization, and dysregulation, was evidenced in the hippocampus and amygdala of AD patients [59]. In addition, aluminum (Al^3+^) is also associated with oxidative stress leading to AD [60].

### 8.4. Inflammation

Neuro-inflammation is expected in sites of metal deposition. Neuro-inflammation is followed by the production of reactive compounds by microglial cells. The microglia can be activated by several factors, including Aβ. Activated microglia play a role in AD onset by increased ROS burden and oxidative stress [61].

### 8.5. Anti-Oxidant

Glutathione, uric acid, vitamin C and E, or antioxidant enzymes (superoxide dismutase, catalase, etc.) are lower in AD patients [30,62]. Failure of the antioxidant defense systems induces OS that is facilitated by Aβ depositions in AD mice with the APP mutation [30].

## 9. Evaluation of ROS in Tauopathies

Since OS is related to the pathophysiological mechanism in tauopathies (Figure 4), several surrogate markers for ROS have been assessed. The measuring parameters mainly cover peroxides including nucleic acid (8-OGH and 8-OHdG, protein or lipid (malondialdehyde, MDA or 4-hydroxynonenal, 4-HNE), antioxidant enzymes, superoxide dismutase (cytosolic SOD1 or mitochondrial SOD2), glutathione levels, glutathione S transferase (GST), heme Oxygenase-1 (HO-1), homocysteine (Hcy) levels, F2-isoprostanes (F2-IsoPs) and vitamins (A, C, or E). These markers are elevated or depleted in particular brain regions of the same patient in tauopathies. A table (Table 1) is provided here based on the available literature for potential OS markers in assessing tauopathies.

## 10. Fluorescence Probes for ROS Detection

Fluorescent probes have advantages for high sensitivity and selectivity over other techniques. Combined with fluorescent microscopy, the fluorescent probes can be used to visualize analytes in living cells spatiotemporally. Also, this technique is fast and straightforward, without a unique facility or high cost-efficiency. Fluorescent sensors (organic fluorophores), when recognizing the target molecule, tend to change their fluorescence emission intensity or color [94,95,96,97]. Here we would like to emphasize some of the basic concepts of the ROS probing based on the fluorescent technique (Table 2).

Fluorescent probes for hydrogen peroxide are mainly based on boronate esters via the H_2_O_2_-induced oxidation of arylboronate ester to phenols [98,99,100]. It is known that under mild alkaline conditions H_2_O_2_ reacts with arylboronic acids and arylboronate ester to produce phenols [101].

Most of the probes for the superoxide radical are based on reaction-based mechanisms, where the superoxide will react with a probe to generate a new molecule with a different fluorescence intensity or color of the probe. These probes are based on superoxide-induced oxidation reactions and reactions with nitroxide [102]. Another essential feature of the superoxide is strong nucleophilicity, where it is known as a super nucleophile. Due to this inherent property, fluorescent probes for superoxide are mainly based on nucleophilic substitution reaction [103]. In this case, most of the time, the leaving group is from the phosphinate group [104], triflate group [105], 2,4-dinitrobenzenesulfonyl [106], nitro-ethers [107], and related groups. Recently, a sulfinate-based chemosensor for superoxide sensing was reported to reduce the molecular size, with good solubility and a cLogP value of 3.8 [108].

Detection of the hydroxyl radical is very challenging due to its low concentration. It is the most reactive among the various classes of the ROS. The hydroxyl radical can oxidize most of the biomolecules like carbohydrates, proteins, and nucleic acids. One of the main classes for hydroxyl radical detection is the oxidation of leuco forms of fluorescent dyes (cyanine, 2,7-dichlorodihydrofluorescein, and dihydrorhodamine) mediated by hydroxyl radical. These probes become highly fluorescent upon reaction with a hydroxyl radical by recovering the π-conjugation networks through hydrogen atom abstraction, and subsequently one-electron oxidation [109,110,111,112]. Some of the nitroxides have been exploited as fluorescent probes for hydroxyl radicals. Here, the diamagnetization reaction between nitroxide and radical restores the fluorescence intensity was exploited, where the CH_3_ radical (generated from hydroxyl radical and dimethyl sulfoxide, DMSO) reacts with the nitroxide moiety to produce the diamagnetic adduct [95,113].

Singlet oxygen is produced by energy transfer from O_2_. As a result, singlet oxygen has paired electrons with opposite spin (spin-restriction state) which enhance oxidizing ability when compared with molecular oxygen. The fluorescent probes for singlet oxygen have been developed based on [2 + 4] cycloaddition, where singlet oxygen acts as a strong dienophile [109,114,115].

Most of the probes for peroxynitrite are based on the organic reactions of probes with peroxynitrite. These reactions include the oxidation of chalcogenides, boronic acids or boronates, hydrazides, the cleavage of C-C double bonds, and oxidative N-dearylation [116,117]. Among these, the most common reaction is the oxidation of chalcogenides (S, Se, and Te) [118,119].

Different types of functional groups have been studied for the detection of NO, which are o-diamino aromatic compounds, luminescent lanthanide complexes, transition-metal complexes, quantum dots, and carbon nanotube sensors. The most commonly studied method for the detection of NO is based on the reaction of a O-diaminophenyl group with NO to generate triazole [120,121].

The fluorescent probes for hypochlorite are mostly based on xanthene probes. The mechanism behind this sensing is the spiro-ring opening of the xanthene probes, which react with hypochlorite [95]. Also, some of the probes are based on the oxidation of catechol to benzoquinone [122].

Apart from these probes, some of the fluorescent probes for the detection of ROS are reported in Alzheimer’s disease. An oxalate-curcumin-based probe was reported for the imaging of reactive oxygen species in AD. Here, the oxalate moiety was utilized, which reacts with H_2_O_2_ to produce the fluorescence signal with a shift in wavelength [123,124,125,126]. A bifunctional fluorescent probe based on the benzothiazole core has been reported for H_2_O_2_ and amyloid aggregate detection. The probe is a combination of benzothiazole and a boronate ester [127].

## 11. Controlling of ROS as Therapeutic Approaches of Tauopathies

Despite the unmet needs for the treatment of tauopathies, FDA-approved treatments are limited to alleviate memory deficits and behavioral changes. Despite the fact that extensive drug discovery programs have been conducted over previous decades, there are still no convincing drugs. Some drugs have adverse side effects or exhibit a lack of cognitive improvement in trial participants. For these reasons, new strategies for therapeutic pathways are being considered. Potent pieces of evidence for the imbalance phenomena of oxidants/antioxidants in AD led to the hypothesis that compounds scavenging free radicals or upregulating the OS defense mechanism might provide therapeutic approaches for AD (Table 3). 

### 11.1. Antioxidant Pathway

By reviewing the OS theories in tauopathies, several therapeutic approaches have been conducted in different tauopathies. Curcumin, found in turmeric, is an antioxidant that decreases Aβ-induced tau hyperphosphorylation in PC12 cells. It also protects PC12 cells through the inhibition of OS [131]. Another class of antioxidant, methylene blue (methylthioninium chloride, MB), which can penetrate the blood brain barrier, diminished oxidized nucleic acids and tau hyperphosphorylation in tau^P301S^ transgenic mice [132]. Administration of coenzyme Q10 (CoQ10; a critical member in electron transport chain), which has antioxidant effects, can significantly reduce tau phosphorylation, lipid peroxidation and OS, while ameliorating behavioral deficits and the survival rate of tau^P301S^ transgenic mice [133]. The overexpression of thioredoxin peroxidase, an antioxidant enzyme, worsens disease phenotypes of tau^R406W^ transgenic drosophila [66]. In PD mice, sulforaphane (found in cruciferous vegetables) were found to be effective at protecting dopaminergic neurons and enhancing GSH levels [134]. Brilliant blue G, an antagonist of the P2X7R pathway of several neurodegeneration, can cross the blood brain barrier to ameliorate neuropathology in AD and PD mouse models [135].

### 11.2. Catalase

Catalase, located in peroxisomes, cytoplasm, and mitochondria, is responsible for H_2_O_2_ conversion into water and oxygen by using iron or manganese as a cofactor. The administration of SOD/catalase mimetic EUK-207 deteriorates disease phenotypes by reducing phosphorylated tau and lipid peroxidation in 3xTg-AD mice [136].

### 11.3. Vitamin

Cells and animal studies, as well as clinical studies, have shown a particular connection between vitamins and tauopathies. The antioxidant activities of vitamins may be useful for the treatment of tauopathies. Proper supplementation of vitamins can reduce the tauopathy incidence in the general population and improve the state of patients. Vitamin C is a water-soluble antioxidant that is abundant in vegetables, fruits, and animal livers. It involves in the inhibition of OS, reducing lipid peroxidation, the exclusion of free radicals, and acts as a cofactor for antioxidant enzymes [137]. Vitamin E is a lipid-soluble high antioxidant which can diminish the effects of peroxide, also protecting against lipid peroxidation in cell membranes. Vitamin C (ascorbic acid), when used to treat primary corticohippocampal neurons from tau transgenic rats (Tau^151–391^ 4R), decreased ROS levels in the neurons, tau inclusions in the spinal cord, and improved behavior [138]. Supplementation of vitamin E (α-tocopherol) delayed tau pathology by attenuating motor weakness in the tau mouse model (B6D2/F1-Tau44) [139]. A recent study has shown that omega-3 fatty acids plus vitamin E supplementation can improve total antioxidant capacity and GSH levels in PD patients [140].

### 11.4. Metal Chelator

Alterations in metallostasis (mainly Zn, Cu and Fe) are responsible for AD. So, ionophore addition would be another treatment choice by restoring ion balances. Clioquinol (5-chloro-7-iodo-quinoline-8-ol) is a moderate chelator for Cu, Zn, and Fe that rescued memory impairment in the TgCRND8-AD mouse after oral administration [141]. The second generation of clioquinol is PBT2, which is now in a clinical trial (phase II) [142]. Both of their modes of action probably rely on copper ionophore activity. Based on this mechanism, another copper containing chelator, Cu^II^GTSM [bis(thiosemicarbazone)], was developed, which could change GSK-3β activity, tau phosphorylation, and restore cognitive impairment in APP/PS1 transgenic mice [143]. One Fe chelator, desferrioxamine, was intramuscularly injected into AD patients, slowing the clinical progression of dementia [144].

## 12. Conclusions

Oxidative stress contributes to the development of tauopathies. It forms vicious pathophysiology inducing mitochondrial dysfunctions, neuronal damages and promotes metal toxicity. The complex nature, genesis, and responses of ROS in tauopathies are still actively being investigated. At the cellular level, individual cells generate excessive levels of ROS, followed by a dysfunctional state of mitochondria that interacts with redox metals and oxidative stress-responsive elements. Although the formation of ROS poses a threat to tauopathies, compensatory responses provoked by ROS removal or the prevention of the ROS generation pathway may interrupt the onset or slow down the progression of tauopathies through multiple mechanisms. These mechanisms include the reduction of oxidative stress-mediated neuronal toxicity, a decrease of tau phosphorylation and aggregation, restoration of mitochondrial function, and metal homeostasis. Therefore, treatment with antioxidants could be an alternative approach to target molecular events implicated in tauopathies. However, evidence on antioxidants as potential therapeutic agents for tauopathies has not been carried out to a significant level. Hence, a more in-depth understanding of OS in tauopathies is in high demanded.

## Figures and Tables

**Figure 1 ijms-20-01959-f001:**
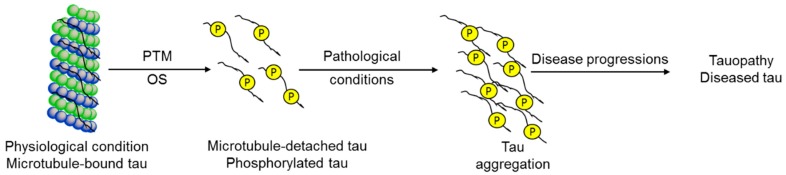
Oxidative stress-mediated tauopathy.

**Figure 2 ijms-20-01959-f002:**
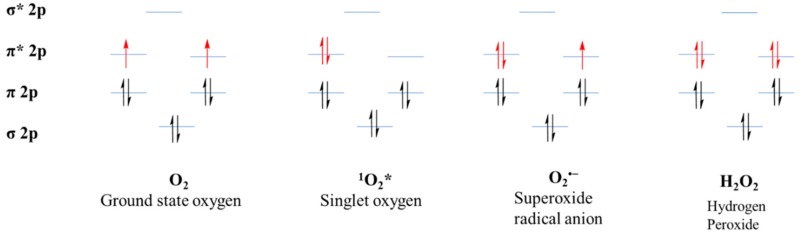
Electronic configuration of reactive oxygen species (ROS).

**Figure 3 ijms-20-01959-f003:**
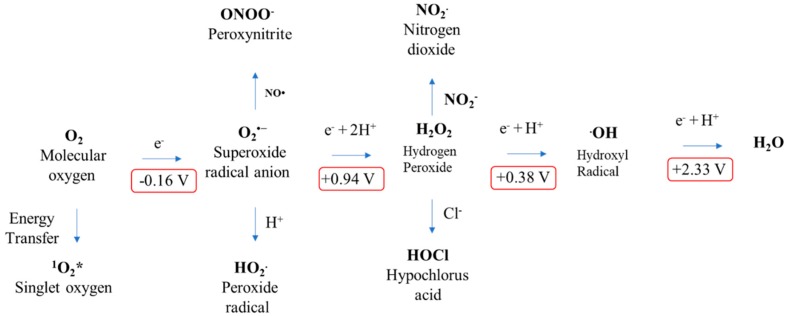
Standard reduction potentials of ROS formation via energy or electron transfer reactions of molecular oxygen.

**Figure 4 ijms-20-01959-f004:**
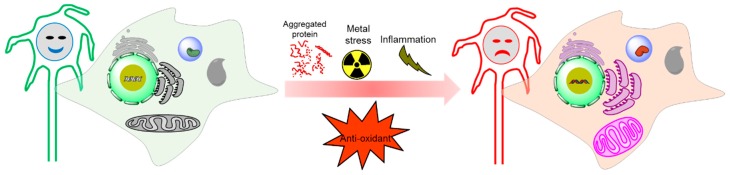
Physiological condition versus tauopathy condition by oxidative stress generation.

**Table 1 ijms-20-01959-t001:** Potential oxidative markers in tauopathies (up-regulation is labeled as up-arrow, and down-regulation is labeled as down-arrow).

OS Markers	AD	PSP	PiD	CBD	FTD	FTDP-17	FTLD	PD
**SOD1**	↓ [27] ↑ [63]	↑ [11]					↑ [64]	↑ [63]
**SOD2**	↓ [27] ↑ [65]	↑ [11]				↓ [66]	↑ [64]	↑ [67]
**Cu/Zn-SOD**	↓ [27] ↑ [68]	↑ [11]						↑ [63]
**GSH**	↓ [27]	↑ [69] ↓ [70]					↓ [64]	↓ [69]
**GST**	↓ [71]					↓ [11]		
**Catalase**	↓ [72]							↓ [73]
**GPx**	↓ [72] ↑ [74]	↑ [75]				↑ [11] ↓ [11]		↑ [76] ↓ [67]
**Lipid peroxidase**	↑ [27]	↑ [77]			↑ [78]			↑ [79]
**MDA**	↑ [27]	↑ [80]					↑ [64]	↑ [81]
**4-HNE**	↑ [27]	↑ [75]	↑ [81]				↑ [64]	↑ [82]
**Vitamin A**	↓ [83]							
**Vitamin C**	↓ [84]							↓ [73]
**Vitamin E**	↓ [85]							↓ [73]
**HO-1**	↑ [86]	↑ [86]	↑ [86]	↑ [86]	↑ [78]			↑ [87]
**Protein carbonyl**	↑ [27]				↑ [78]			↑ [63]
**3-nitrotyrosine**	↑ [27]							↑ [88]
**8-OHG**	↑ [27]							↑ [89]
**8-OHdG**	↑ [27]							↑ [89]
**F2-IsoPs**	↑ [27]							
**Hcy**	↑ [90]							
**COX**	↑ [91]							
**PDHC**	↓ [92]							
**KGDHC**	↓ [93]							

**Table 2 ijms-20-01959-t002:** ROS and their mode of detection using fluorescent probes.

ROS	Fluorescent Probe	Reaction	Examples
**Hydrogen peroxide**	H_2_O_2_-induced oxidation of arylboronate ester to phenols.	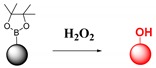	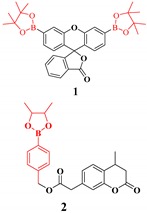 (Ref: 1 [98], 2 [99])
**Superoxide**	Nucleophilic substitution reaction of superoxide with the probes where the leaving group is phosphinate group [104], triflate group [105], 2,4-dinitrobenzenesulfonyl [106], nitro-ethers [107] and the related groups.	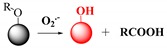	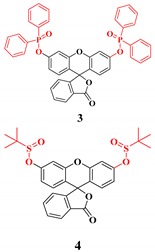 (Ref: 3 [128],4 [108])
**Hydroxyl radical**	Hydroxyl radical-mediated oxidation of leuco forms of fluorescent dyes and nitroxide moiety.	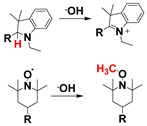	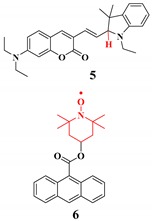 (Ref: 5 [112], 6 [129])
**Singlet oxygen**	The fluorescent probes for singlet oxygen have been developed based on the [2 + 4] cycloaddition where singlet oxygen acts as a strong dienophile [109,114,115].	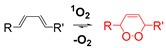	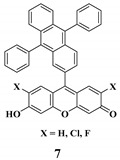 (Ref: 7 [115])
**Peroxynitrite**	Peroxynitrite-mediated oxidation of chalcogenides (S, Se, and Te), boronic acids or boronates, hydrazides, cleavage of C–C double bonds and oxidative *N*-dearylation.	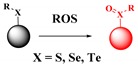	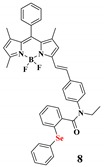 (Ref: 8 [122])
**Nitric oxide**	NO-mediated reaction of O-diaminophenyl group with NO to generate the triazole.	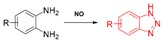	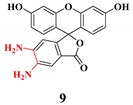 (Ref: 9 [130])
**Hypochlorite**	Hypochlorite mediated spiro-ring opening of the xanthene probes and oxidation of catechol to benzoquinone [122].	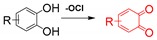	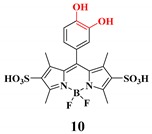 (Ref: 10 [117])

**Table 3 ijms-20-01959-t003:** Therapeutic approaches of tauopathies related to ROS.

Therapeutic Approaches	Chemical Agents	Treating Route and Doses	Experimental Model	Reference
**Antioxidant pathway**	Curcumin	10 μg/mL for 1 h	Aβ treated PC12 cell	[131]
	Methylene blue	4 mg/kg in diet	Tau^P301S^ mouse	[132]
	CoQ10	0.5% in diet	Tau^P301S^ mouse	[133]
	Paraquant	30 mM in diet for 48 h	Tau^R406W^ drosophila	[66]
	Sulforaphane	5 mg/kg twice a week by intraperitoneally	C57Bl/6^6-OHDA^ mouse	[134]
	BR 297	500 nM for 24 h	APP treated SH-SY5Y cell	[145]
	S14	5 mg/Kg/daily for 4 weeks by intraperitoneally	Tg2576^APP/PS1^ AD mouse	[146]
	Resveratrol	500–1000 mg/daily for 26 months by orally	AD patients	[147]
**Catalase**	EUK-207	3.41 mM/day for 28 days by micro-osmotic pump	C57BL/6/129S-3xTg-AD mouse	[136]
**Vitamin**	Ascorbic acid	250–500 μM for 24 h	Neurons form Tau^151–391^ rat	[138]
	α-tocopherol	0.5–1.5 mM in diet for 10 days	tau^R406W^ drosophila	[66]
		49 IU/Kg in diet	B6D2/F1-tau44 mouse	[139]
		2000 IU/day for 6 months	AD patient	[148]
	Tocotrienol	5 mg/Kg/day for 15 months by orally	APPswe/PS1dE9 mouse	[149]
**Metal chelator**	Clioquinol	30 mg/kg/day for 5 weeks by orally	TgCRND8-AD mouse	[141]
	PBT2	250 mg/day for 12 weeks by orally	AD patient	[142]
	Cu^II^GTSM	10 mg/kg/daily by orally	APP/PS1 AD mouse	[143]
	Desferrioxamine	125 mg twice daily/5 days per week for 24 months by intramuscularly	AD patient	[144]

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
