# Peer review of "Crosstalk between Oxidative Stress and Tauopathy"

_ijms, 2019, doi:10.3390/ijms20081959_

Round 1
Reviewer 1 Report
Dear Authors,
This a great review related to the role of oxidative stress in tauopathy. Before the full acceptance of the manuscript I would to suggest the following changes:
- Include an abbreviations list.
- I would remove the section “fluorescence probes for ROS detection” and instead develop more in deep “evaluation of ROS in tauopathies”.
- Expand the section “controlling of ros as therapeutic approaches of tauopathies” and correlate, if possible with the previous section (evaluation of ros in tauopathies).
Thank you very much.
Author Response
Response to Reviewer 1.
Dear Authors,
This a great review related to the role of oxidative stress in tauopathy. Before the full acceptance of the manuscript I would to suggest the following changes:
- Include an abbreviations list
Ans:Thank you for the comments regarding the abbreviations. The list is added in the text before references.
- I would remove the section “fluorescence probes for ROS detection” and instead develop more in deep “evaluation of ROS in tauopathies”.
Ans:First of all, we truly appreciate your valuable concerns. We believe the table showing fluorescent probes will be of great interests among biologist who got interested in to know more about the chemical structures and their mode of action with ROS. Therefore, we added more descriptions in section “fluorescence probes for ROS detection”. Based on the reviewer’s comment, we have added more ROS markers for tauopathies along with the corresponding references in the section of “evaluation of ROS in tauopathies.”
- Expand the section “controlling of ros as therapeutic approaches of tauopathies” and correlate, if possible with the previous section (evaluation of ros in tauopathies).
Ans:As per the reviewer’s comment, we expanded the section “controlling of ros as therapeutic approaches of tauopathies”. In this section, we have explained the detail of the therapeutic approaches with the recent references.
Thank you very much.

Reviewer 2 Report
The present review entitled "Crosstalk between oxidative stress and tauopathy" by Haque et al. represents a tentative to summarize the novelty about the chemical nature of the reactive oxygen species and their role in tauopathy, but unfortunately failed to do the same.
I have some concerns about the manuscript. In my opinion, the manuscript needs to be enhanced for its publication in the International Journal of Molecular Science.
1. The manuscript should be better organized. Authors describe deeply oxidative stress and ROS properties and its generation and detection, but the “crosstalk” between them and tauopathies is barely described. In my opinion authors should investigate more deeply the role of ROS production with more examples from the literature.
2. The paragraph 11 is not exhaustive. I would suggest to discuss more about the potential anti-oxidative approaches. Maybe authors should include also a table to summarize the mentioned studies with the tested molecule and the in vitro or in vivo model used.
3. The literature used in the review is not recent. Authors should use the more recent literature and if not available, it means that no progress has been made in the field and therefore no review is necessary.
4. Authors should include in the Table 1 the references.
5. The Table 2 is too long. Authors should use tables as a summary and introduce all the information in the main text.
6. The language of the manuscript is so poor with many typographical errors, grammatical errors and inappropriate sentences.
7. The phrase at line 81-82 makes no sense.
Author Response
Response to Reviewer 2.
The present review entitled "Crosstalk between oxidative stress and tauopathy" by Haque et al. represents a tentative to summarize the novelty about the chemical nature of the reactive oxygen species and their role in tauopathy, but unfortunately failed to do the same.
I have some concerns about the manuscript. In my opinion, the manuscript needs to be enhanced for its publication in the International Journal of Molecular Science.
1. The manuscript should be better organized. Authors describe deeply oxidative stress and ROS properties and its generation and detection, but the “crosstalk” between them and tauopathies is barely described. In my opinion authors should investigate more deeply the role of ROS production with more examples from the literature.
Ans:Thank you for your vital comments. As per the reviewer’s opinion, we have explained the section “oxidative stress and its relation to tauopathies” in detail with more references. This revised manuscript has explained how ROS is produced in the tauopathy as well as how ROS is mediated the tauopathies progression. This is also explained a bit in the section entitled “ROS production in the body” and “ROS generating agents in tauopathies”.
2. The paragraph 11 is not exhaustive. I would suggest to discuss more about the potential anti-oxidative approaches. Maybe authors should include also a table to summarize the mentioned studies with the tested molecule and the in vitro or in vivo model used.
Ans:As per the reviewer’s suggestion, we have included the potential approaches (antioxidant pathway, vitamin, metal chelator etc) in paragraph 11. We have added a table mentioning the dose of the tested compound, treating route, and experimental cells, or animals or patients.
3. The literature used in the review is not recent. Authors should use the more recent literature and if not available, it means that no progress has been made in the field and therefore no review is necessary.
Ans:As per the reviewer’s comment, the literature lists are updated with the recent publications.
4. Authors should include in the Table 1 the references.
Ans: The references are added in Table 1.
5. The Table 2 is too long. Authors should use tables as a summary and introduce all the information in the main text.
Ans:Table 2 is edited to make it short and comprehensive more. The information is introduced in the main text under the title “fluorescence probes for ROS detection”.
6. The language of the manuscript is so poor with many typographical errors, grammatical errors and inappropriate sentences.
Ans:We have revised the manuscript more carefully to overcome those typo and grammatical errors. The inappropriate sentences are deleted from this revised version.
7. The phrase at line 81-82 makes no sense.
Ans:As per the reviewer’s comment we have changed the sentence for clarification. Therefore, we re-wrote those sentences to improve readability as following. “The endogenous sources of OS are (i) the mitochondrial electron transport chain (ETC); the factors which interruptthe ETC arehypoxia, NO and ONOO− that endorse the generation of ROS in the mitochondria, (ii) increased level of misfolded proteins, (iii) the intracellular enzymes that produce oxidants (ROS) because of their enzymatic processes;these enzymes include NADPH oxidases, flavoenzyme ERO1, cytochrome p450, lipoxygenases, xanthine oxidase, nitric oxide synthases etc; thesemetal ions include copperand iron ions”.

Round 2
Reviewer 1 Report
You address all of my comments and concerns. Congratulations for the article!
Author Response
Thank you very much for your positive encouragement. We truly believe that this article will appeal broad readership from chemist to brain scientists.
Reviewer 2 Report
Authors have improved their manuscript. Anyway, there are still some typographical and grammatical errors. Just to highlight some of them, please check line 79-80, line 110, line 118-122, line 277-278.
Author Response
Thank you very much for the valuable and fruitful comments. Once again, we carefully revised the manuscript and revised accordingly as the reviewer suggested. We believe after this revision, the manuscript improves a lot in terms of readability. We appreciate the comments from the reviewer.